# One Instruction Is a Benchmark: End-to-End Instruction-Following Evaluation with FlexBench

## Abstract

As Large Language Models (LLMs) grow increasingly capable and accept ever longer, more nuanced instructions, models exhibit wide variability in their instruction following across prompts. Fixed, general-purpose benchmarks therefore fail to reflect real deployment performance, where evaluation must be tailored to the instruction at hand. We present **FlexBench**, a self-evolving framework that automatically constructs a specialized instruction-following benchmark comprising a set of evaluation dimensions and a conversation corpus from only a *single instruction*. For evaluation, we also introduce **FlexEval**, which aggregates tri-valued (yes/no/unknown), per-dimension decisions into instruction-level metrics that jointly capture workflow progress *(Coverage)* and conditional correctness *(Achievement)*. Together, our work establishes a fully automated paradigm for customized benchmark construction, which enables instruction-specialized evaluations that adapt to arbitrary task requirements and deliver fine-grained, reproducible judgments of instruction following, turning open-ended instructions into end-to-end evaluations. We validate our framework on 248 single-turn complex instructions, and further conduct extensive experiments on 10 leading LLMs across three multi-turn conversational scenarios with complex instructions. Our results show that FlexBench and FlexEval deliver instruction-specialized assessments and provide actionable insights for improving LLM instruction following. Our source code is publicly available at `https://anonymous.4open.science/r/FlexBench-E0D7`

## 1 Introduction

Large Language Models (LLMs) have made rapid progress across generation, reasoning, and problem solving (Achiam et al., 2023; Chowdhery et al., 2023; Brown et al., 2020). As their capabilities grow, so do users' expectations: real deployments increasingly rely on long, precise, and sometimes idiosyncratic instructions. Yet, instruction-following remains fragile. Models that excel on broad, general-purpose leaderboards can still fail to restate content verbatim, obey formatting constraints, respect conditional logic, or carry out multi-step workflows when these behaviors are demanded by a particular instruction. Crucially, a model's "general" ability is not a reliable proxy for its ability to follow a specific instruction. This gap is amplified as instructions lengthen and compound requirements. As evidenced in Figure 1, mainstream LLMs—including Claude (Anthropic, 2025), Gemini 2.5 Flash (Comanici et al., 2025), Qwen3-32B (Yang et al., 2025), Doubao 1.5 Pro (ByteDance, 2025), and GPT4 (OpenAI, 2025) variants—exhibit marked performance disparities when evaluated across three instructions of different settings(V1, V2, and V3). These models demonstrate substantial variance across both achievement (overall task correctness) and coverage (step-wise completeness), underscoring a pronounced model-specific sensitivity to variations in instruction structure, length, and semantic type. This inconsistency reveals a critical shortcoming of prevailing static evaluation paradigms: their inability to disentangle instruction-form effects from intrinsic model capabilities, thereby limiting their validity in instruction-specific settings. These observations directly motivate our framework, FlexBench, which explicitly incorporates instruction heterogeneity through automated dimension derivation and leakage-free conversation synthesis—ensuring robust and targeted assessment of instruction-following behavior.

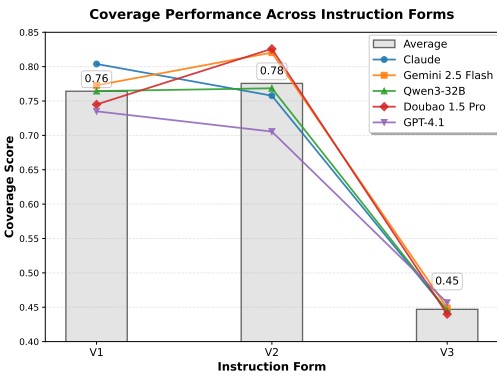 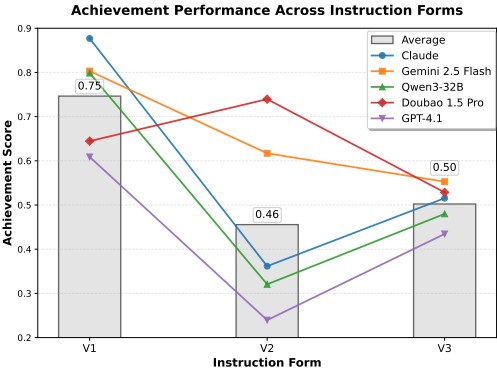

(a) Coverage across instruction forms.

(b) Achievement across instruction forms.

Figure 1: Across three instruction forms, mainstream LLMs (e.g., Claude, Gemini 2.5 Flash, Qwen3-32B, Doubao 1.5 Pro, GPT4.1) exhibit non-uniform performance patterns in both achievement and coverage, indicating model-specific sensitivity to instruction length/type/form. This motivates our design to explicitly model instruction-form heterogeneity.

Meanwhile, existing benchmarks offer valuable signals but rely on fixed datasets and lack scenario-, task-, or prompt-specific construction. For example, (Qin et al., 2024) introduces instruction decomposition, but its dataset is limited, manual, and unavailable for very long instructions, and its metric is a flat ratio without logical aggregation. (Wen et al., 2024) defines logic-aware dimensions, but the logical structures are manually extracted rather than induced automatically. (Zeng et al., 2023) simulates conversational data, but only for single-turn settings. In contrast, our framework constructs benchmarks *on demand* from a single user instruction, is fully automated end-to-end without human curation, and performs logic-aware aggregation to separately capture workflow coverage and conditional correctness.

We introduce **FlexBench**, a self-evolving framework that constructs a specialized benchmark $\mathcal{B}(I) = (\mathcal{D}, \mathcal{C})$ from a *single* seed instruction $I$. As formalized in Eq. 1, FlexBench maps $I$ to a set of evaluation dimensions $\mathcal{D}$ and to a conversation corpus $\mathcal{C}$, during which dimensions are derived automatically. Conversations are synthesized by a user simulator that preserves an instruction-invariant backbone and injects only a minimal, instruction-specific profile delta, explicitly *excluding* evaluation dimensions to prevent leakage during rollout.

On top of these components, we instantiate **FlexEval**, a conditional aggregator that consumes only per-dimension, tri-valued outcomes to produce two complementary metrics. *Coverage* measures workflow progress by crediting attempts on base actions regardless of correctness (Eq. 6), reflecting whether the model advances all required steps instead of skipping them. *Achievement* measures conditional correctness on gated branches, i.e., "once the model acts, how often is it right?" (Eq. 7). This design separates "doing" from "doing correctly," avoids dependence on free-form judges, and respects preconditions via principled gating and mutual-exclusion rules.

**Contributions.**

- **Self-evolving, instruction-targeted benchmarking.** We propose *FlexBench*, which constructs a complete benchmark from a single instruction: a fine-grained dimension set and a conversation corpus. The process is fully automated and requires no existing datasets or manual curation.

- **Automatic dimension derivation with optional preconditions.** We introduce an instruction-level factorization pipeline that yields entity-explicit, non-redundant dimensions.

- **Leakage-free conversation generation.** We design a user simulator with an instruction-invariant backbone and a constrained profile delta, ensuring stylistic adaptation without exposing evaluation dimensions to the simulator.

- **Conditional aggregation for workflow completeness and correctness.** We instantiate FLEX-EVAL to compute *Coverage* and *Achievement* from only tri-valued per-dimension outcomes, enabling reliable, instruction-conditioned evaluation without external judges.

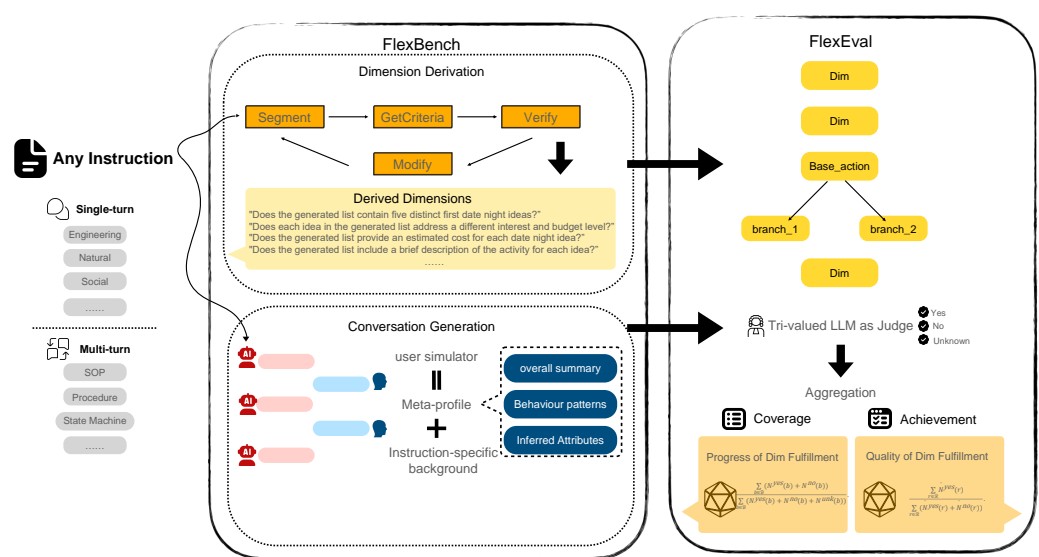

Figure 2: Workflow of FlexBench and FlexEval.

## 2 RELATED WORK

Instruction-following evaluation has been studied by many researchers. IFEval (Zhou et al., 2023) introduced verifiable instructions, and InfoBench (Qin et al., 2024) decomposed complex requirements into atomic constraints. ComplexBench (Wen et al., 2024) explored compositional constraints, QA-IF (Adlakha et al., 2024) studied correctness vs. faithfulness in factual QA, LIFBench (Wu et al., 2024) tested stability under long contexts, and StrucText-Eval (Gu et al., 2024) examined reasoning on structured text. Reliability and robustness have been addressed by LLM-BAR (Zeng et al., 2023) and Automatic Dialogue Evaluators (Zhang et al., 2024), while self-evolving benchmarks (Wang et al., 2024) aim to mitigate staleness. Broader benchmarks such as BIG-bench (Srivastava et al., 2023), HELM (Liang et al., 2022), MT-Bench (Bai et al., 2024), and AlpacaEval (Li et al., 2023), along with resources like Super-NaturalInstructions (Wang et al., 2022), Tülu (Wang et al., 2023), Arena-Hard (Li et al., 2024), Chatbot Arena (Chiang et al., 2024), and AlpacaFarm (Dubois et al., 2023), further contribute large-scale or community-driven evaluations.

Domain-specific work extends these directions. CodeIF-Bench (Wang et al., 2025) targets interactive code generation, MathIF (Fu et al., 2025) evaluates reasoning models under compositional mathematical constraints, and InfoSearch (Zhou et al., 2024) and IFIR (Song et al., 2025) benchmark retrieval with nuanced or expert-domain instructions. While these highlight important facets of instruction following, limitations persist: handcrafted taxonomies, dataset dependence, evaluator bias, and narrow domains. FlexBench moves beyond these by inducing benchmarks directly from input instructions, ensuring coverage, long-range dependency, evaluability, and faithful maximality in a fully automated and scalable framework.

## 3 FLEXBENCH: A SELF-EVOLVING FRAMEWORK

As shown in Figure 2, we formalize benchmark construction from a single instruction. Given a seed instruction $I$, **FlexBench** produces a specialized instruction-following benchmark $\mathcal{B}(I)$ consisting of an evaluation-dimension set and a conversation corpus:

$$I \xrightarrow{\text{FlexBench}} (\mathcal{D}, \mathcal{C}), \qquad \mathcal{D} = \{d_i\}_{i=1}^{D}, \;\; \mathcal{C} = \{c_j\}_{j=1}^{M}. \tag{1}$$

In this setup, the instruction $I$ specifies the task requirements and is shared across all generated artifacts. The dimension set $\mathcal{D}$ enumerates the dimensions along which instruction-following will be evaluated (cardinality $D = |\mathcal{D}|$). The conversation corpus $\mathcal{C}$ comprises single-turn and multi-turn

conversations (size $M = |\mathcal{C}|$), each constructed to operationalize $I$ and to elicit model behavior that completes the tasks specified by $I$.

**FlexBench** constructs a specialized evaluation benchmark through a fully automated two-stage pipeline that requires no human intervention:

1. **Dimension Extraction.** The system first analyzes the target instruction $I$ and derives a complete set of evaluation dimensions $\mathcal{D} = \{d_i\}_{i=1}^{D}$ through automatic decomposition.

2. **Conversation Generation.** Based on the extracted dimensions, the framework then synthesizes diverse conversational sequences $\mathcal{C} = \{c_j\}_{j=1}^{M}$, incorporating both single-turn and multi-turn interaction patterns.

The resulting benchmark $\mathcal{B}(I) = (\mathcal{D}, \mathcal{C})$ provides a comprehensive testbed for assessing instruction-following behavior specific to the provided instruction $I$.

### 3.1 DIMENSION DERIVATION

To construct evaluation dimensions from an instruction $I$, we begin by parsing $I$ into a finite set of requirement clauses $\mathrm{Spec}(I) := \{r_1, \ldots, r_m\}$, where each clause $r_j$ represents a distinct aspect of the instruction's specification. Our objective is to derive a set of evaluation dimensions $\mathcal{D} = \{d_i\}_{i=1}^{D}$ that provides complete coverage of instruction $I$ through the mapping $\Gamma : \mathrm{Spec}(I) \to \mathcal{D}$.

To properly evaluate each dimension, we operate over conversation traces $\tau \in \mathcal{T}$, where $\mathcal{T}$ represents the space of all possible multi-turn interaction sequences. For each dimension $d_i$, we define an evaluation predicate $R_i : \mathcal{T} \to \mathcal{V}$ that assesses whether the dimension's requirement is satisfied, where the evaluation codomain $\mathcal{V} = \{\mathrm{yes}, \mathrm{no}, \mathrm{unknown}\}$ corresponds to: *yes* indicating successful satisfaction of the dimension, *no* indicating explicit violation, and *unknown* representing inconclusive evidence.

For dimensions requiring contextual activation, we define precondition functions $A_i : \mathcal{T} \to \{0, 1\}$, where $A_i(\tau) = 1$ indicates all necessary contextual conditions are met for evaluating dimension $d_i$ on trace $\tau$, and $A_i(\tau) = 0$ indicates otherwise. Let $\mathcal{A} \subseteq \{1, \ldots, D\}$ denote the indices of dimensions equipped with such preconditions.

The evaluation rule for dimension $d_i$ on trace $\tau$ is then defined as:

$$s_{d_i}(\tau) := \begin{cases} \mathrm{unknown}, & \text{if } i \in \mathcal{A} \wedge A_i(\tau) = 0 \\ R_i(\tau), & \text{otherwise} \end{cases} \tag{2}$$

This formulation ensures that dimensions are evaluated only when their contextual preconditions are satisfied, while properly handling cases where evaluation evidence remains inconclusive. The complete evaluation across all dimensions for a given conversation trace $\tau$ is represented as a judgment vector:

$$\mathbf{s}_{\mathcal{D}}(\tau) := \left(s_{d_1}(\tau), \ldots, s_{d_D}(\tau)\right) \in \mathcal{V}^D, \tag{3}$$

which provides a comprehensive assessment of instruction compliance along each derived dimension. These dimension-level verdicts are then aggregated across all conversation traces in $\mathcal{C}(I)$ through an instruction-specific aggregator:

$$\Phi_I^{\mathrm{FLEXEVAL}} : \left(\mathcal{V}^D\right)^{|\mathcal{C}(I)|} \to \mathcal{M}, \tag{4}$$

where $\mathcal{M}$ denotes the space of evaluation metrics.

**Operational Pipeline.** Therefore, the derivation process follows a structured pipeline: (1) Instruction Segmentation, (2) Decontextualization and Coreference Canonicalization, (3) Verification, and (4) Modification. Concrete implementation details and prompting strategies are provided, see Appendix B for more detailed workflow and prompt details.

We then implement $\Phi_I^{\mathrm{FLEXEVAL}}$ using the conditional aggregation scheme detailed in Section 4, which produces two interpretable metrics:

$$\Phi_I^{\mathrm{FLEXEVAL}} \left(\{\mathbf{s}_{\mathcal{D}}(\tau)\}_{\tau \in \mathcal{C}(I)}\right) = (\mathrm{Coverage}(I), \mathrm{Achievement}(I)) \in [0, 1]^2.$$

These metrics are computed through principled gating and counting mechanisms that properly account for precondition dependencies and evidence uncertainty across dimensions.

**Design Guarantees.** A dimension set $\mathcal{D}$ derived from instruction $I$ is **valid** if it satisfies:

**(i) Coverage Completeness.** The decomposed dimensions collectively span all requirements in $I$, so instruction-level evaluation is determined entirely by dimension-level decisions.
**(ii) Long-range Dependence.** Each dimension's judgment depends only on the semantics of its referenced fragments (a subset of $\{r_1, \ldots, r_m\}$) and remains invariant to meaning-preserving reorderings, faithful paraphrases, or non-altering insertions elsewhere in $I$.
**(iii) Evaluability.** Each dimension is entity-explicit and verifiable from conversation traces; preconditions are directly checkable, and predicates may return `unknown` when evidence is insufficient.

## 3.2 Conversation Generation

Given an instruction $I$, we synthesize a diverse conversation corpus $\mathcal{C}(I) = \{c_j\}_{j=1}^{M}$ through a user simulation approach that combines a general base persona with lightweight instruction-specific adaptations. This design ensures naturalistic interactions while preventing evaluation leakage. Persona acquisition details and JSON examples are deferred to the Appendix D.

**User Simulation Framework.** We first define a taxonomy of five user personas characterized by varying levels of cooperativeness. All personas share a domain-agnostic base prompt $P_0$ containing stable attributes including personality traits, turn-taking conventions, and communication protocols. These core attributes remain consistent across all instructions to maintain behavioral coherence.

**Instruction-Specific Adaptation.** To preserve evaluation integrity, we deliberately avoid exposing evaluation dimensions to the simulator. Instead, we infer a minimal set of stylistic profile fields that naturally correlate with instruction content—specifically common catchphrases and behavioral patterns. This approach preserves the behavioral backbone while allowing surface-level adaptations to instruction content.

**Conversation Rollout.** For each conversation $c_j$, we instantiate a user simulator $U_{P_I}$ with the adapted profile and interact with the model under test $M$ using unique random seed $z_j$:

$$c_j \sim \text{Rollout}\big(M, U_{P_I}; z_j\big), \quad \mathcal{C}(I) = \{c_j\}_{j=1}^{M} \tag{5}$$

Rollouts terminate upon detection of task completion, simulator-initiated termination (due to uncooperativeness), or upon reaching a maximum turn limit $T_{\max}$. We employ temperature sampling and seed variation to encourage behavioral diversity while maintaining profile consistency.

**Design Guarantees.** Our generation framework provides three key guarantees:
**(i) Evaluation Integrity:** Neither $P_0$ nor $\Delta(I)$ contains evaluation dimensions or their templates, preventing the simulator from strategically influencing the evaluation.
**(ii) Controlled Variability:** The system maintains an instruction-invariant core persona while allowing limited, semantically-grounded adaptations through well-defined profile fields.
**(iii) Collusion Resistance:** The absence of few-shot examples in prompting prevents imitation artifacts and reduces the risk of evaluation collusion.

## 4 FlexEval: Conditional Aggregation of Dimension Results

This section specifies how tri-valued, per-dimension verdicts are aggregated into instruction-level metrics under optional preconditions. We emphasize two complementary aspects of instruction following: *workflow completeness* and *conditional correctness*, captured by Coverage and Achievement.

### 4.1 Setup: conditional groups and tri-valued outcomes

Let $\mathcal{D}$ be the dimensions derived in Section 3.1, and $\mathcal{C}(I)$ the set of conversations generated in Section 3.2 for instruction $I$. Each dimension produces a tri-valued verdict (yes, no, unknown) interpreted as success, failure, or insufficient evidence/inapplicability.

To operationalize preconditions, we employ a large language model to extract a *tree-structured logic* over the dimension set, resulting in a forest of precondition groups $\mathcal{G} = \{g_k\}_{k=1}^{K}$. Each group $g \in \mathcal{G}$ consists of:

- A single base dimension $d_g^{\text{base}} \in \mathcal{D}$ (annotated as `base_action`)
- A potentially empty set of dependent dimensions $\mathcal{R}_g \subseteq \mathcal{D}$ (annotated as `branch_k`)

Dimensions without preconditions are represented as singleton groups with $\mathcal{R}_g = \emptyset$. The aggregation mechanism remains agnostic to the specific extraction methodology; prompt templates and concrete examples are provided in the Appendix.

## 4.2 METRICS

For each dimension $d \in \mathcal{D}$, we define corpus-level summary statistics over $\mathcal{C}(I)$ as:

$$N^{(v)}(d) = \sum_{\tau \in \mathcal{C}(I)} \mathbb{I}\left[s_d(\tau) = v\right], \quad \text{for } v \in \{\text{yes}, \text{no}, \text{unknown}\}.$$

**Coverage (workflow progress).** Coverage quantifies the model's ability to *advance the workflow and fully express required base steps*. Intuitively, higher Coverage means fewer omissions: when an operation is required, the model attempts it rather than skipping.

$$\text{Coverage}(I) := \frac{\sum_{b \in \mathcal{B}} \left(N^{\text{yes}}(b) + N^{\text{no}}(b)\right)}{\sum_{b \in \mathcal{B}} \left(N^{\text{yes}}(b) + N^{\text{no}}(b) + N^{\text{unk}}(b)\right)}. \tag{6}$$

**Achievement (conditional correctness).** Achievement measures *accuracy conditional on acting*: among operations the model chooses to perform (bases affirmed), what fraction of dependent checks are correct—i.e., "once the model acts, how often is it right?"

$$\text{Achievement}(I) := \frac{\sum_{r \in \mathcal{R}} \tilde{N}^{\text{yes}}(r)}{\sum_{r \in \mathcal{R}} \left(\tilde{N}^{\text{yes}}(r) + \tilde{N}^{\text{no}}(r)\right)}. \tag{7}$$

If the denominator is zero for an instruction (no gated branches), we report "n/a".

## 4.3 AGGREGATION LOGIC

**Mutual-exclusion rule (workflow progress).** *Coverage* measures workflow progress by crediting attempts regardless of correctness. For independent dimensions, a dimension is covered when it is answered. For mutually exclusive alternatives (an exclusive-choice set), we treat the whole set as a single workflow unit: the unit is covered in a conversation if any one of its alternatives is attempted.

**Gating rule (conditional correctness).** We compute *Achievement* by first enforcing prerequisites: a branch contributes to correctness only if its base action in the same group is affirmed. Otherwise, the branch is treated as not applicable for correctness. This guarantees that dependent checks are evaluated only when the required action was actually taken, preventing spurious credit from unsupported branches.

## 5 EXPERIMENTS

For experiments, the following models were utilized: Claude (Claude Sonnet 4) (Anthropic, 2025), Gemini 2.5 Flash, Gemini 2.5 Flash Lite (Comanici et al., 2025), Qwen3-32b (Yang et al., 2025), Doubao 1.5 Pro (Doubao 1.5 Pro 32k), Doubao 1.5 Role (Doubao 1.5 Pro 32k character 0715), Doubao 1.5 Lite (Doubao 1.5 Lite 32k) (ByteDance, 2025), GPT (GPT 4.1), GPT Mini (GPT 4.1 Mini), and GPT Nano (GPT 4.1 Nano) (OpenAI, 2025).To avoid randomness during the decoding process while preserving the variety of datasets, we use temperature=0.3 for all the models.

Table 1: **Overall evaluation results.** Table 1a reports semantic alignment, and Table 1b shows LLM-as-judge comparisons.

(a) **Semantic alignment** between OURS and HUMAN across 1,560 aligned pairs.

| Metric | Mean | Std |
|---|---|---|
| BERT cos. (sent.) | 0.928 | 0.044 |
| BERTScore–P | 0.820 | 0.092 |
| BERTScore–R | 0.819 | 0.085 |
| BERTScore–F1 | 0.818 | 0.082 |
| ROUGE-L | 0.635 | 0.171 |

(b) **LLM-as-judge comparison** between OURS (A) and HUMAN (B) from 248 instructions.

| Criterion | A win (%; $\Delta$) | B win (%) | Ties (%) |
|---|---|---|---|
| Overall (N=248) | 62.1 (+24.6) | 37.5 | 0.4 |
| Granularity | 61.7 (+33.9) | 27.8 | 10.5 |
| Eval. Suitability | 62.1 (+25.8) | 36.3 | 1.6 |
| Completeness | 56.5 (+26.3) | 30.2 | 13.3 |
| Coverage | 54.8 (+25.0) | 29.8 | 15.3 |
| Task Alignment | 29.8 (+17.3) | 12.5 | 57.7 |
| Clarity | 25.4 (−14.1) | 39.5 | 35.1 |

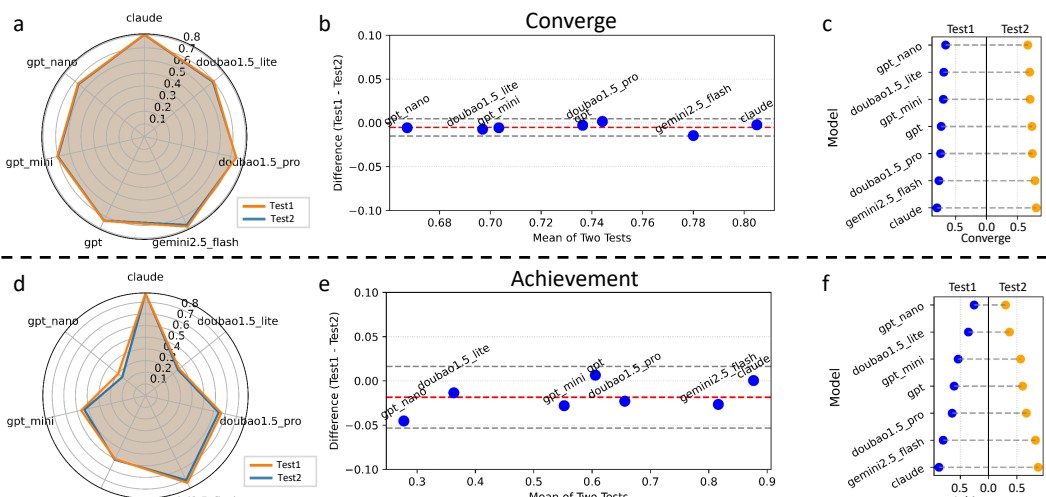

Figure 3: Consistency analysis for converge (a-c) and achievement (d-f) across seven models in two tests. (a, d) Radar charts comparing model coverage and achievement across test sets. (b, e) Bland-Altman plots showing agreement between Test1 and Test2. (c, f) Pair line plots illustrating cross-model changes between Test1 and Test2, indicating that no crossovers occur and the relative ranking of models is preserved. In the pair line plots, the scores (coverage or achievement) in each test are sorted by value, and dashed lines connect results from the same model.

## 5.1 DECOMPOSITION

**Setup.** We compare automatic dimension derivation (OURS) with human-authored decompositions (HUMAN) on **248** instructions using two protocols: (i) *LLM-as-judge* pairwise preferences over six criteria (Granularity, Evaluation Suitability, Completeness, Coverage, Task Alignment, Clarity); and (ii) *semantic alignment* on **1,560** aligned pairs via sentence-level BERT cosine, token-level BERTScore (P/R/F1), and ROUGE-L F1.

**Results.** Semantic alignment indicates strong content overlap: BERT cosine 0.928, BERTScore F1 0.818, ROUGE-L F1 0.635. LLM-as-judge prefers OURS in 154/248 cases (62.1%), with large gains on *Granularity* (+33.9 pts) and *Evaluation Suitability* (+25.8 pts); only *Clarity* favors HUMAN (−14.1 pts). Overall, OURS maintains content fidelity while producing more evaluable, task-aligned decompositions, validating faithful coverage and logic-aware assessment. Moreover, OURS perform even better among disimilar pairs, see C for more comparisons.

Table 2: **Main results (Top-10 models) across instruction types (3–5).** Using the *provided* overall *Coverage* and *Achievement* from the dataset (higher is better).

| Model | C.(d1) | A.(d1) | C.(d2) | A.(d2) | C.(d3) | A.(d3) | C.(d4) | A.(d4) | C.(d5) | A.(d5) | C. ↑ | A. ↑ |
|---|---|---|---|---|---|---|---|---|---|---|---|---|
| Claude | 1.000 | 1.000 | 0.900 | 1.000 | 1.000 | 0.972 | 1.000 | 0.467 | 0.920 | 1.000 | **0.804** | **0.877** |
| Gemini 2.5 Flash | 0.800 | 0.769 | 0.900 | 0.895 | 0.800 | 1.000 | 1.000 | 0.745 | 0.880 | 0.932 | 0.773 | 0.803 |
| Qwen3-32B | 0.760 | 0.857 | 0.960 | 1.000 | 0.760 | 0.968 | 0.880 | 0.583 | 0.880 | 0.955 | 0.764 | 0.799 |
| Doubao 1.5 Pro | 0.480 | 0.700 | 0.360 | 0.750 | 0.480 | 0.929 | 1.000 | 0.086 | 0.740 | 0.946 | 0.745 | 0.645 |
| GPT-4.1 | 0.760 | 0.938 | 0.640 | 0.231 | 0.760 | 0.500 | 0.500 | 0.150 | 0.760 | 1.000 | 0.735 | 0.609 |
| Gemini 2.5 Flash Lite | 0.840 | 0.957 | 0.560 | 0.158 | 0.840 | 1.000 | 0.500 | 0.467 | 0.520 | 1.000 | 0.745 | 0.565 |
| Doubao 1.5 Role | 0.660 | 1.000 | 0.500 | 0.529 | 0.660 | 0.500 | 0.500 | 0.067 | 0.780 | 1.000 | 0.731 | 0.557 |
| GPT-4.1-mini | 0.320 | 0.778 | 0.340 | 0.067 | 0.320 | 0.429 | 0.740 | 0.050 | 0.280 | 1.000 | 0.701 | 0.538 |
| Doubao 1.5 Lite | 0.040 | 0.000 | 0.240 | 0.100 | 0.040 | 0.500 | 0.200 | 0.000 | 0.380 | 0.684 | 0.693 | 0.356 |
| GPT-4.1-nano | 0.040 | 0.000 | 0.100 | 0.000 | 0.040 | 0.000 | 0.000 | 0.000 | 0.000 | 0.000 | 0.664 | 0.255 |

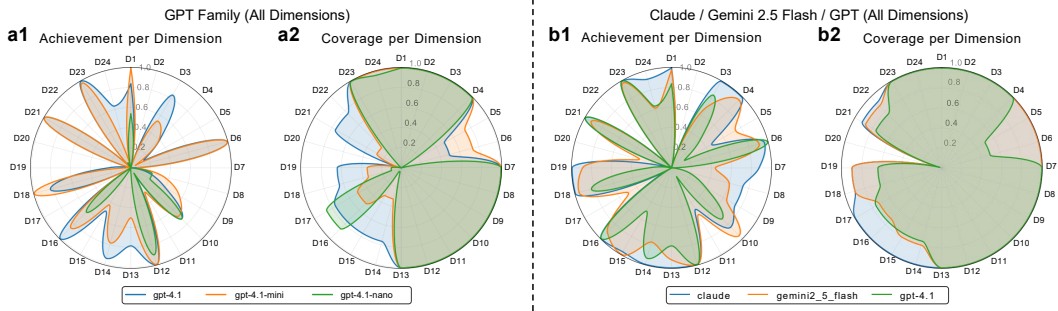

Figure 4: Comparison of different models across 24 dimensions. Five models are evaluated in total. (a) shows comparisons within the GPT family, with (a1) and (a2) presenting achievement and coverage, respectively. (b) compares Claude, Gemini 2.5 Flash, and GPT-4.1, with (b1) and (b2) presenting achievement and coverage, respectively.

## 5.2 SELF AND CROSS-MODEL CONSISTENCY

To examine the stability of our benchmark construction methods, we simulate two independent dialogue datasets generated from the same instruction and evaluate seven models on both. We compare coverage and achievement across the two tests to assess consistency between datasets. The average absolute error, expressed as $Mean \pm Std.$, is $(0.0056 \pm 0.0041)$ for coverage and $(0.0204 \pm 0.0139)$ for achievement.

Figure 3 summarizes the results of two tests across seven models. Figures 3(a) and (d) show that coverage and achievement in Tests 1 and 2 are similar overall. A more detailed comparison is presented using two complementary methods. Figures 3(b) and (e) present Bland-Altman plots (BA plots), indicating that differences across tests remain small and stable for all models, which demonstrates high self-consistency. Figures 3(c) and (f) show pair line plots where results from each test are sorted by value and connected across models. No crossings are observed, meaning the relative performance ranking of models remains unchanged, which demonstrates high consistency and stability.

## 5.3 MAIN RESULTS

Table 2 and Figure 4 summarize model performance. **Claude** achieves the best overall results (Coverage = 0.804, Achievement = 0.877), followed by **Gemini 2.5 Flash** and **Qwen3–32B**. **GPT-4.1** reaches competitive Coverage (0.735) but lower Achievement (0.609), showing that it attempts most steps yet is less reliable on conditional branches. Across instruction types, $d4$ consistently depresses Achievement (e.g., Claude 0.467, GPT-4.1 0.150), highlighting the difficulty of logic-gated or formatting-sensitive requirements.

Radar plots reveal consistent family trends: stronger models show smoother profiles with fewer sharp drops. Importantly, the **GPT family follows scaling laws**, with GPT-4.1 outperforming its mini and nano variants in both metrics, especially on dimensions requiring multi-step reason-

Table 3: Dimension-level accuracy (Overall Accuracy; correct if $\arg\max$ prediction matches expert label per dialogue and dimension).

| Model | Crowd-sourced | LLM-as-Judge | Dialogues | Dims |
|---|---|---|---|---|
| `claude` | 0.611 | 0.883 | 21 | 504 |
| `doubao1_5_pro` | 0.852 | 0.798 | 20 | 480 |
| `gemini2_5_flash` | 0.772 | 0.888 | 13 | 312 |
| `gpt` | 0.782 | 0.852 | 18 | 432 |
| **Micro avg (dims)** | 0.750 | 0.852 | 72 | 1,728 |
| **Macro avg (models)** | 0.755 | 0.855 | – | – |

Table 4: Cross-model Pearson correlations $r$ with expert annotations (system-level consistency).

| Pair | Coverage $r$ | Achievement $r$ |
|---|---|---|
| Crowd-sourced vs Ground Truth | 0.043 | -0.227 |
| LLM-as-Judge vs Ground Truth | 0.997 | 0.986 |

ing. Cross-family comparisons confirm that Coverage saturates quickly, while Achievement differentiates models more clearly—supporting our claim that both metrics are necessary to capture instruction-following ability.

### 5.4 PEARSON CORRELATION

**Results and Analysis.** Table 3 shows that LLM-as-judge achieves higher dimension-level accuracy than crowd-sourced annotations (micro average 0.852 vs. 0.750, macro average 0.855 vs. 0.755), with consistent improvements on most models. At the system level (Table 4), correlations with expert annotations are nearly perfect ($r = 0.997$ for *Coverage*, $r = 0.986$ for *Achievement*), whereas crowd-sourced labels correlate weakly or negatively. Overall, LLM-as-judge not only aligns more closely with expert judgments but also preserves relative model ranking, making it a more reliable evaluation signal for subsequent experiments.

## 6 CONCLUSION AND FUTURE WORK

We introduced **FlexBench**, a self-evolving benchmark constructed automatically from a single seed instruction. Unlike prior work based on fixed datasets or manual decomposition, FlexBench is fully automated end-to-end, requiring no human-written rubrics or annotations. Every benchmark instance is induced directly from the instruction itself, enabling task- and scenario-specific evaluation rather than relying on static task banks. Built on this foundation, **FlexEval** aggregates tri-valued, per-dimension outcomes into two complementary metrics: *Coverage*, capturing whether models attempt the required steps, and *Achievement*, capturing whether they act correctly when they do. This separation of workflow completeness from conditional correctness provides a robust and interpretable view of instruction-following performance.

FlexBench demonstrates that instruction-specialized, self-evolving evaluation is both feasible and necessary as LLMs are increasingly deployed on long, structured, and consequential tasks. While limitations remain—such as handling extremely long or underspecified prompts, inducing rare logical dependencies, and extending beyond text-only traces—our framework establishes a foundation for instruction-targeted benchmarking that can scale with real-world demands. Future work will extend FlexBench to richer modalities and strengthen automatic reasoning over dependencies, further advancing reliable and adaptive evaluation for the next generation of LLMs.

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

## A DETAILS IN INSTRUCTION SELECTION

### A.1 SINGLE-TURN

For the single-turn setting, we directly adopt the `Hard Set` from InfoBench (Qin et al., 2024). The Hard Set is manually curated by experts and covers **72 domains** ranging from Natural Sciences, Social Sciences, Engineering, Economics, Arts, to daily-life tasks (see Figure 1 in the original paper). Compared with the Easy Set, the Hard Set features **longer instructions** (average length $\sim$59 words), **more requirements per instruction** (average $\sim$6.3), and **richer constraint types** (Content, Linguistic, Style, Format, and Number). This makes it significantly more challenging and effective in distinguishing model capabilities, particularly for complex instruction-following. We use the Hard Set as the gold standard, comparing its human-provided decompositions with our automatically extracted dimensions.

### A.2 MULTI-TURN

For the multi-turn setting, we construct three domain-specific instructions covering education services, cross-border e-commerce logistics, and on-demand delivery platforms: (i) *Customer Support Specialist for a Course Publishing Platform*, (ii) *Cross-Border 3PL Fulfillment Caller Assistant*, and (iii) *Station Leader for FastRun Riders*. Figure 5 illustrates the *Customer Support Specialist* instruction, which requires informing institutional clients about new streaming options ("Standard Live" vs. "Low-Latency Direct") while following strict procedures. These characteristics highlight why multi-turn settings are a more realistic and challenging evaluation scenario.

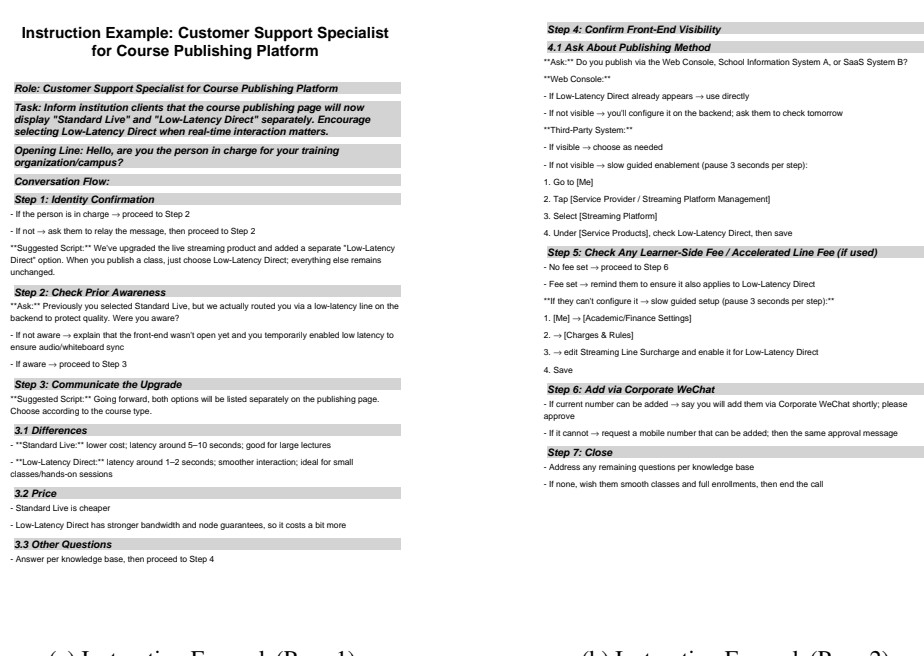

(a) Instruction Example(Page 1).    (b) Instruction Example(Page 2).

Figure 5: Multi-turn instruction example: Customer Support Specialist for Course Publishing Platform.

Table 5 summarizes the statistics of our three multi-turn instructions. Compared to the single-turn Hard Set (average length $\sim$59 words), our multi-turn instructions are on average much longer ($\sim$258 words, $4.4\times$), contain more sentences ($\sim$12), and additionally embed state-machine style preconditions and branching flows. Compared to single-turn cases, our multi-turn instructions are significantly more complex, and substantially harder for humans to decompose. In particular, the presence of preconditions and branching flows makes manual decomposition nearly impossible.

| Instruction | Words | Sentences |
|---|---|---|
| Customer Support Specialist | 490 | 19 |
| Cross-Border 3PL Assistant | 140 | 6 |
| Station Leader (FastRun) | 145 | 11 |
| **Average** | **258.3** | **12.0** |

Table 5: Length and sentence statistics of multi-turn instructions.

## B DIMENSION EXTRACTION PROMPTS & DETAILED END-TO-END PIPELINE

This appendix details our semi-automatic pipeline that converts a fully-specified instruction into a structured set of conversation-observable dimensions and a sharded (segment-level) equivalent suited for evaluation. The procedure follows a four-stage LLM-driven process—**Segmentation**, **Criteria Generation**, **Coverage Verification**, and **Targeted Modification**—augmented with two engineering enhancements for **Gradual** reduction and **Dependency** consolidation. All core prompts used in the LLM-driven stages are provided in the appendix (see Prompts A–D in Figures 6–9).

### B.1 SEGMENTATION

For this procedure, we follow (Laban et al., 2025) segmentation prompt to get robust segments. Given a fully-specified instruction, we first segment it into *non-overlapping*, *conversation-observable* units of information ("segments") that preserve the instruction's core behavioral requirements while avoiding excessive fragmentation. The prompt explicitly enforces non-overlap, encourages minimal yet meaningful granularity, and prioritizes segments that can be verified through actual multi-turn dialogue.

### B.2 CRITERIA GENERATION

Using the original instruction together with the segments, we convert each segment into a dimension stated as an *evaluation question* anchored to an observable output artifact. The prompt enforces *zero hallucination and strict grounding* to the instruction/segments, requires explicit *conditional dependencies* ("if.../when..."), and applies a *summary + specific script* dual structure whenever recommended dialogue scripts exist (capturing behavioral intent without inventing constraints).

### B.3 VERIFICATION

We then assess whether the dimension set adequately covers the core information in the original instruction. The verifier returns `{"coverage":"complete"}` if coverage is sufficient; otherwise it returns `{"coverage":"incomplete","missing_segment":...}` and pinpoints the critical missing unit. This diagnostic signal drives the next step's targeted revision.

### B.4 MODIFICATION

When coverage is incomplete, we perform localized edits based on the verification feedback while preserving the original JSON schema. The modification prompt adds missing dimensions, clarifies vague criteria, completes conditional logic, and improves measurability, but avoids scope creep or unconstrained expansion. The revised set is re-verified (Step 3) until coverage is complete.

### B.5 IMPLEMENTATION DETAILS

Our implementation is fully automated and executes the pipeline as: *Segmentation → Criteria → Verification → (Modification)\* → Gradual → Dependency\** (\* applied as needed). Key practicalities include:

- **Fully Automated without Human Intervene.** The entire pipeline is executed automatically without requiring any human input at intermediate steps. Each stage (segmenta-

tion, criteria generation, coverage verification and modification) is driven by LLM prompts (Prompts A–D) and programmatic wrappers, ensuring that raw instructions are transformed into structured dimension sets end-to-end. Robust JSON repair mechanisms, retry logic, and verification–modification loops are incorporated to guarantee stability and completeness, thereby eliminating the need for manual correction during the process.

- **Retry with temperature annealing.** On verification failure or malformed outputs, we trigger bounded retries with slight temperature increases to promote diversity while maintaining determinism.

- **Coverage-first control flow.** Only dimension sets verified as complete (Step 3) proceed to Gradual/Dependency refinements; otherwise we loop through Targeted Modification (Step 4) and re-verify.

## Prompt A — Segmentation (Instruction → Segments)

```
You are given a fully specified instruction for a conversational AI assistant, and your task is to segmen

**Important Context**: These segments will be used to generate evaluation criteria for assessing whether

You must output a list of segments in the following JSON format:
[
    {"segment": "[exact excerpt from the instruction]"},
    {"segment": "[exact excerpt from the instruction]"},
    ...
]

Rules:
- [Non-overlapping] The segments must be non-overlapping and cover the entire instruction. You can option
- [Minimalistic] Split information into units, but maintain meaningful groupings. If an expression includ
- [No excessive splitting] Avoid breaking down compound expressions into individual units unless it adds
- [Conversation-focused] Focus on segments that describe observable behaviors, communication patterns, or

Example Query:
What are the names and locations of the stadiums that had concerts that occurred in both 2014 and 2015?

Output:
{"segments": [
    {"segment": "names and locations"},
    {"segment": "stadiums"},
    {"segment": "concerts"},
    {"segment": "in both 2014"},
    {"segment": "and 2015"}
]}

Now complete the task for the following fully specified instruction:

[[INSTRUCTION]]
```

Figure 6: Prompt A: Segmentation (Instruction → Segments)

## Prompt B — Criteria (Evaluation Question Generation)

```
You are given segments of a conversational AI instruction. Your task is to:
1. Convert each segment into an evaluation question that assesses observable conversation behaviors.
2. For segments with conditional logic, include explicit dependency conditions using "if..." or "when..."
3. **Handle recommended scripts carefully** - preserve key details while maintaining clarity.

**ZERO-HALLUCINATION & GROUNDING RULES (CRITICAL):**
- All criteria MUST be directly grounded in the provided "Instruction" and "Segments".
- DO NOT introduce any new entities, settings, roles, numbers, limits, formats, or constraints that are n
- Only paraphrase; do not expand scope. If the input does not specify a quantity/limit (e.g., "no more th
- If a segment is broad or ambiguous, write the dimension as an observable action without adding extra sp
- Do not carry constraints across segments unless the dependency is explicitly stated. Each dimension mus

**Core Guidelines:**
- Focus on **observable actions** in conversations, not knowledge testing.
- **Replace step references with concrete action descriptions**: Instead of "proceed to the next step" or
- **Emphasize complex dependency conditions**: For multi-conditional logic, make dependencies crystal cle
- Use specific, measurable criteria ONLY IF such metrics are explicitly present in the source text. Other
- Avoid vague conditions like "in relevant situations".
- Do not infer missing details such as locations, counts, personas, or formatting if not provided.

**OUTPUT ANCHORING RULE (CRITICAL):**
- **R1 - Anchor to Output Artifact**: Always reference "the generated [specific artifact]" (e.g., "the ge

**Script Handling - SUMMARY + DETAIL APPROACH:**
- **IMPORTANT DISTINCTION**: Only apply to **conversational scripts/dialogue content**, NOT to format/con
- **What qualifies as a "script"**: Specific dialogue content, suggested wording, recommended phrases for
- **What does NOT qualify**: Format requirements (e.g., "exactly 10 words", "JSON format"), length constr
- **Two-part structure for recommended scripts**: First summarize the core intent, then include the speci
- **Format**: "Does the generated [artifact] [summary of the script's intent], with the specific recommen
- **CRITICAL: Intelligent Summarization Rules**:
  - **DO NOT repeat the script content in the summary.**
  - **Extract the BEHAVIORAL INTENT or COMMUNICATION PURPOSE.**
  - **Use different words to describe the ACTION, not the content.**
  - **Focus on WHAT the agent is trying to achieve, not WHAT they say.**
- **Avoid Redundancy**: The summary should capture the essence/intent, while the script provides the exac
- Only apply this when a recommended script is actually present. Do NOT fabricate scripts.
- **Examples of WHAT QUALIFIES as Scripts (use Script Handling)**:
  - ✓ "The discount can be as low as 60%, but not every order gets a 60% discount." → Summary: "clarify
  - ✓ "I'm very sorry, perhaps my information was incorrect. My apologies for bothering you, please hang
  - ✓ "We apologize, but we have made many improvements now. Would you like to try it again?" → Summary:
  - ✓ "The shipping subsidy is not issued as a coupon; the system will automatically apply a discount to

- **Examples of WHAT DOES NOT QUALIFY as Scripts (direct evaluation, no Script Handling)**:
  - ✗ "exactly 10 words" → Direct: "Does the generated response contain exactly 10 words?"
  - ✗ "no more than 3 sentences" → Direct: "Does the generated response contain no more than 3 sentences
  - ✗ "JSON format" → Direct: "Is the generated response in valid JSON format?"
- **Then attach full script**: Include the complete recommended script for verification purposes.
- **Balance completeness with clarity**: Ensure both summary and detail are present.

**Dependency Condition Requirements:**
- **Multi-step dependencies**: When a dimension depends on complex prior conditions, make ALL conditions
- **Conditional branching**: Clearly specify which condition leads to which specific action.
- **Sequential dependencies**: Make the order and prerequisites crystal clear.
- Do NOT transfer constraints between branches unless explicitly stated.

**Anti-Extrapolation Examples:**
- BAD: Adding "no more than 7 sentences" when no such limit exists in the input.
- BAD: Forcing a "restaurant" setting when the input only says "create a scene".
- GOOD: If the segment says "confirm the user is a merchant, then inform delivery subsidy", write: "When
Your output must be valid JSON only:
```

Figure 7: Prompt B: Criteria Generation (Evaluation Question Generation)

## Prompt C — Verification (Coverage Judgement)

```
You are given an instruction that fully specifies a problem, and a list of dimensions. Your task is to de

The dimensions should cover the essential elements needed to complete the task. Minor details or optional

If the core requirements are not covered, you should output the critical information unit from the instru

Example 1:

Instruction:
What are the names and locations of the stadiums that had concerts that occurred in both 2014 and 2015?

Dimensions:
{"initial_segment": "stadiums", "initial_dimension": "I'm looking for active stadiums", "dimensions": [{"

Output:
{"coverage": "complete"}

Example 2:
Instruction:
Which Asian countries have a population that is larger than any country in Africa?

Dimensions:
{"initial_dimension": "I'm interested in learning about countries in Asia", "dimensions": [{"dimension":

Output:
{"coverage": "incomplete", "missing_segment": "the dimensions do not specify that the population of the A

You must output in JSON format as shown in the examples above.
Now complete the task for the following fully specified instruction and dimensions:

Instruction:
[[QUERY]]

Dimensions:
[[DIMENSIONS]]
```

Figure 8: Prompt C: Verification (Completeness Judgement)

## Prompt D — Modification (Refine Dimensions by Feedback)

You are an expert specialized in modifying and improving conversational dimensions. Your task is to modif

## Input Information:
1. **Original Instruction**: Complete task instruction
2. **Current Dimensions**: Existing dimension structure
3. **Verification Feedback**: Specific information about what is missing or insufficient

## Modification Principles:

### 1. Maintain Structural Integrity
- Keep the original JSON structure format: `{"dimensions": []}`
- The dimensions array should contain all important evaluation points

### 2. Precise Modifications Based on Feedback
- **If feedback indicates missing information**: Add new dimensions to cover the missing content
- **If feedback indicates unclear expression**: Rephrase relevant dimensions to make them clearer
- **If feedback indicates missing logic**: Add necessary conditional dependency relationships
- **If feedback indicates insufficient detail**: Enhance the specificity and measurability of relevant di

### 3. Follow Conversational Evaluation Best Practices
- **Focus on observable behaviors**: Ensure each dimension can be verified through specific behaviors in
- **Clear conditional dependencies**: Use "when..." or "if..." to clearly express conditional relationshi
- **Concrete step references**: Replace abstract "next step" with specific action descriptions
- **Maintain script integrity**: For recommended scripts, use "intent summary + specific script" dual str

### 4. Quality Assurance Principles
- **Completeness**: Ensure coverage of all key information points in the original instruction
- **Accuracy**: Each modified dimension should accurately reflect corresponding instruction requirements
- **Measurability**: Each evaluation criterion should be verifiable through conversation observation
- **Logical clarity**: Conditional branches and dependency relationships should be clearly expressed

## Output Requirements:
- Output the complete modified dimensions JSON structure
- Ensure JSON format is correct and directly parseable
- Output only JSON, no other explanatory text

## Example Modification Scenarios:

**Scenario 1: Missing Key Information**
- Feedback: Missing handling of specific conditions
- Modification: Add new dimensions or supplement missing conditions in existing dimensions

**Scenario 2: Insufficient Specificity**
- Feedback: A certain dimension is vaguely expressed
- Modification: Rephrase that dimension to make it more specific and measurable

**Scenario 3: Missing Dependencies**
- Feedback: Missing logical connections between steps
- Modification: Add clear conditional dependency expressions in relevant dimensions

Now please modify the dimensions based on the following information:

Original Instruction:
[[INSTRUCTION]]

Current Dimensions:
[[CURRENT_DIMENSIONS]]

Verification Feedback:
[[VERIFICATION_FEEDBACK]]

Figure 9: Prompt D: Modification (Refine Dimensions by Feedback)

## C  DECOMPOSITION COMPARISON BETWEEN OURS AND HUMAN

In this section, we provide further analysis comparing the results of dimensions decomposed by humans versus those automatically extracted with our methods.

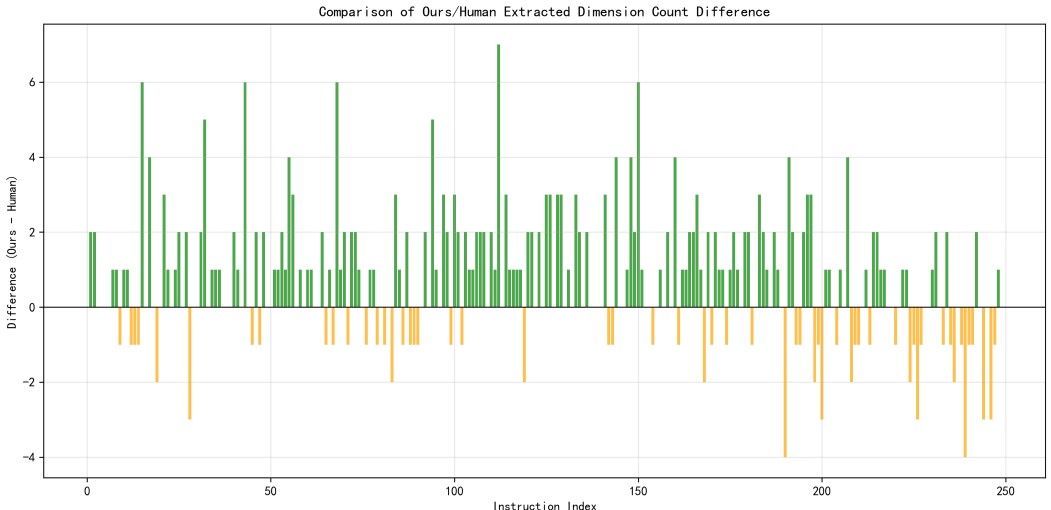

(a) Difference of number of dimensions from Human decomposed and LLM auto-extracted.

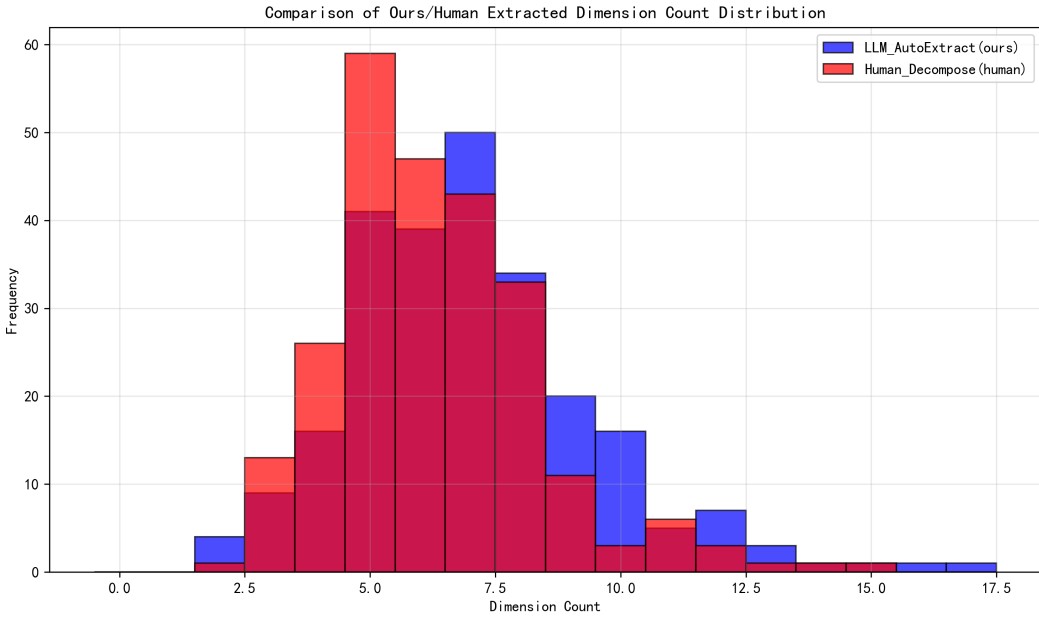

(b) Distribution of number of dimensions from Human decomposed and LLM auto-extracted.

Figure 10: Comparison between human-decomposed and LLM auto-extracted dimensions: (a) difference in number of dimensions, (b) distribution of dimensions.

## C.1 STATISTICS OF DECOMPOSITION RESULTS

We first investigate the number of derived dimensions. Figure 10 presents a comparison between human-decomposed and LLM auto-extracted dimensions. Overall, both approaches yield similar ranges of dimension counts (typically 4–7), suggesting broad alignment. However, the difference analysis shows that the LLM often extracts slightly more dimensions, while the distribution comparison highlights that the LLM produces a heavier tail with higher counts (occasionally exceeding 15), indicating a tendency towards a finer-grained decomposition relative to human annotators.

## C.2 Further Analysis of Dissimilar Pairs

We further investigate the subset of instructions where the sentence-level BERT cosine similarity between human and LLM decompositions falls below 0.85, indicating notable disagreement. For these divergent cases, a closer examination reveals that the LLM-as-judge evaluation consistently favors our LLM-based decomposition. Specifically, our method achieves substantially higher scores across six evaluation dimensions, significantly surpassing the overall average performance. This suggests that when disagreements occur, the LLM-extracted decompositions tend to capture finer-grained and more useful structures than human annotations.

## D Conversation Generation Prompts

The conversation generation module leverages the constructed user profiles to synthesize realistic dialogues under different cooperativeness levels. The process can be summarized as follows:

**Profile-driven user simulation.** We first map each cooperativeness score (1–5) to a corresponding *Meta User Profile*, which encodes communication style, information disclosure preferences, problem–solving approaches, and typical catchphrases. These profiles are then injected into a simulator prompt template, forming the role description for simulated users.

**Dialogue initialization.** Each dialogue starts with the assistant issuing an *opening line* defined in the instruction template. The simulator then generates user responses conditioned on the profile, ensuring that behaviors are consistent with the specified cooperativeness level.

**Turn-taking and control.** Dialogues proceed in alternating turns between the assistant (task-driven agent) and the simulator (profile-driven user). We cap the maximum number of rounds per dialogue to maintain controllability and diversify the interaction lengths.

**End-of-conversation detection.** The system incorporates a hybrid mechanism to decide when a dialogue should terminate. On the one hand, strict keyword rules capture explicit hangup expressions (e.g., "goodbye", "stop calling"). On the other hand, an LLM-based detector jointly evaluates conversation history and the latest user response to identify either (i) user hangup intent or (ii) the assistant reaching its final closing step. A dialogue is ended once either condition is satisfied, with the reason (user termination vs. assistant completion vs. max rounds reached) logged.

### D.1 Meta User Profiles

In this paper, we present only the results of our *Meta User Profiles*, as the categorization of users is not fixed and can be flexibly defined or adapted to the requirements of different application scenarios. Nevertheless, our approach offers a convenient and generalizable framework for constructing conversation datasets with only minor modifications.

We construct user profiles in three stages:

- **(i)** Raw dialogue data are first categorized by cooperativeness levels using an automated grader.
- **(ii)** Dialogues within the same category are aggregated into *Meta User Profiles* (see Figure 11 for details).
- **(iii)** Given downstream instructions, these meta profiles are further adapted into *Adjusted User Profiles*.

### D.2 Adjusted User Profiles

As mentioned in Appendix D.1, we first construct *Meta User Profiles* by aggregating dialogues within the same cooperativeness category. To adapt these category-level profiles to specific downstream tasks, we further generate *Adjusted User Profiles*. The adjustment procedure proceeds as follows:

**Aggregated User Profiles (Meta)**

```
{
  "Poor cooperation": {
    "profile": {
      "cooperativenessCategory": {
        "category": "Poor cooperation"
      },
      "summary": "Users in this category show obvious defensiveness, perfunctory responses, or impatience in co
      "behavioralPatterns": {
        "communicationStyle": "Mainly uses brief, repetitive, questioning, perfunctory, or irrelevant statement
        "informationDisclosure": "Low willingness to disclose information, rarely actively provides information
        "problemSolvingApproach": "Problem-solving approach is mainly avoidance, evasion, and perfunctory respo
      },
      "inferredAttributes": {
        "primaryGoals": [
          "Maintain existing operational habits, avoid additional learning or changes",
          "Save time, reduce irrelevant or lengthy communication, pursue operational simplicity"
        ],
        "commonCatchphrases": [
          "You say (then)",
          "What's the matter (ah/er)?",
          "What happened?",
          "Who is this?",
          "Who are you?",
          "Then what?",
          "Is there something?",
          "Are you a robot?",
          "No need/Don't need/No time/Goodbye",
          "What's the difference?",
          "Has the price increased now?"
        ],
        "impliedPersonalityTraits": [
          "Strong defensiveness, suspicious or distrustful attitude toward external communication",
          "Lack of patience, easily shows impatience or perfunctory responses",
          "Resistant to change, tends to maintain status quo",
          "Passive coping, low initiative",
          "Focus on practical benefits, low interest in details and process optimization"
        ]
      }
    }
  },
  "General cooperation": {
    "profile": {
      "cooperativenessCategory": {
        "category": "General cooperation"
      },
      "summary": "Users in this category show basic willingness to cooperate but have limited initiative and en
      "behavioralPatterns": {
        "communicationStyle": "Communication style is mainly concise and direct, with mostly neutral or slightl
        "informationDisclosure": "Information disclosure is mainly passive. Users mostly provide required infor
        "problemSolvingApproach": "When encountering problems, users tend to first express questions or confusi
      },
      "inferredAttributes": {
        "primaryGoals": [
          "Ensure simple and clear processes, convenient operations",
          "Focus on speed, costs, and service quality, avoid platform changes affecting their own interests"
        ],
        "commonCatchphrases": [
          "How to operate?",
          "What's the difference from before?",
          "Will the price change?",
          "How to calculate overtime?",
          "Are you a robot?",
          "I've always used direct service",
          "You say it",
          "Can/Can't",
          "Say it again"
```

Figure 11: Aggregated Meta User Profiles constructed from categorized dialogue data.

- **Instruction analysis.** Given a new instruction, we use an LLM to extract its business context, key topics, user concerns, technical terms, typical questions, and intended user goals.

- **Behavioral pattern update.** Based on the analysis, we refine the meta profile's communication style, information disclosure tendencies, and problem-solving approaches, inserting instruction-specific examples (e.g., replacing generic platform references with task-relevant business contexts).

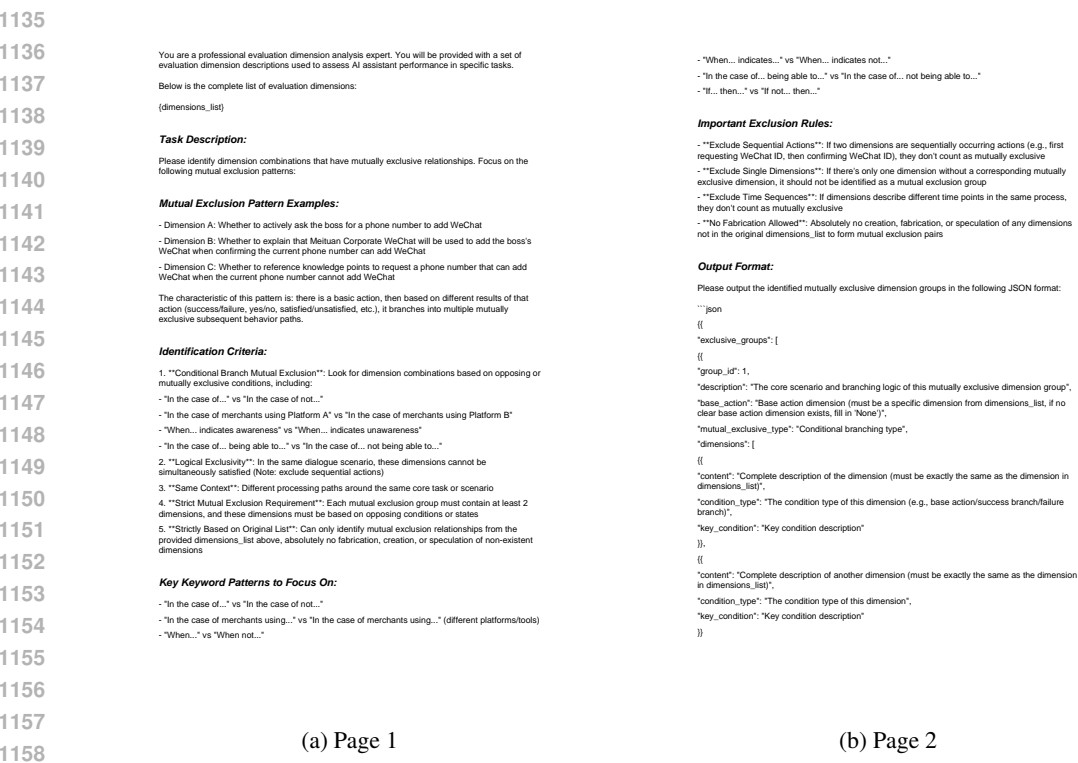

(a) Page 1             (b) Page 2

Figure 12: Exclusive pattern identification template.

- **Catchphrase enrichment.** We augment the profile's common catchphrases by adding 1–2 representative questions or terms derived from the instruction (e.g., frequent user questions about new technical terms).

This procedure ensures that the originally general-purpose *Meta User Profiles* are minimally but effectively adapted to the requirements of each instruction, yielding instruction-specific *Adjusted Profiles* that remain compatible with our simulation pipeline.

# E  LOGIC EXTRACTION PROMPTS

See Figure 12 for our automatic logic extraction prompts. Used to extract a *tree-structured logic* over the derived dimension set, resulting in a forest of precondition groups.

# F  DETAILED EVALUATION RESULTS

See Figure 13 for specific derived dimensions and Figure14 and Figure15 for detailed coverage and achievement for all models among all dimensions.

# G  THE USE OF LARGE LANGUAGE MODELS (LLMS)

Only to aid or polish writing. Certain sentences are translated and polished by LLMs.

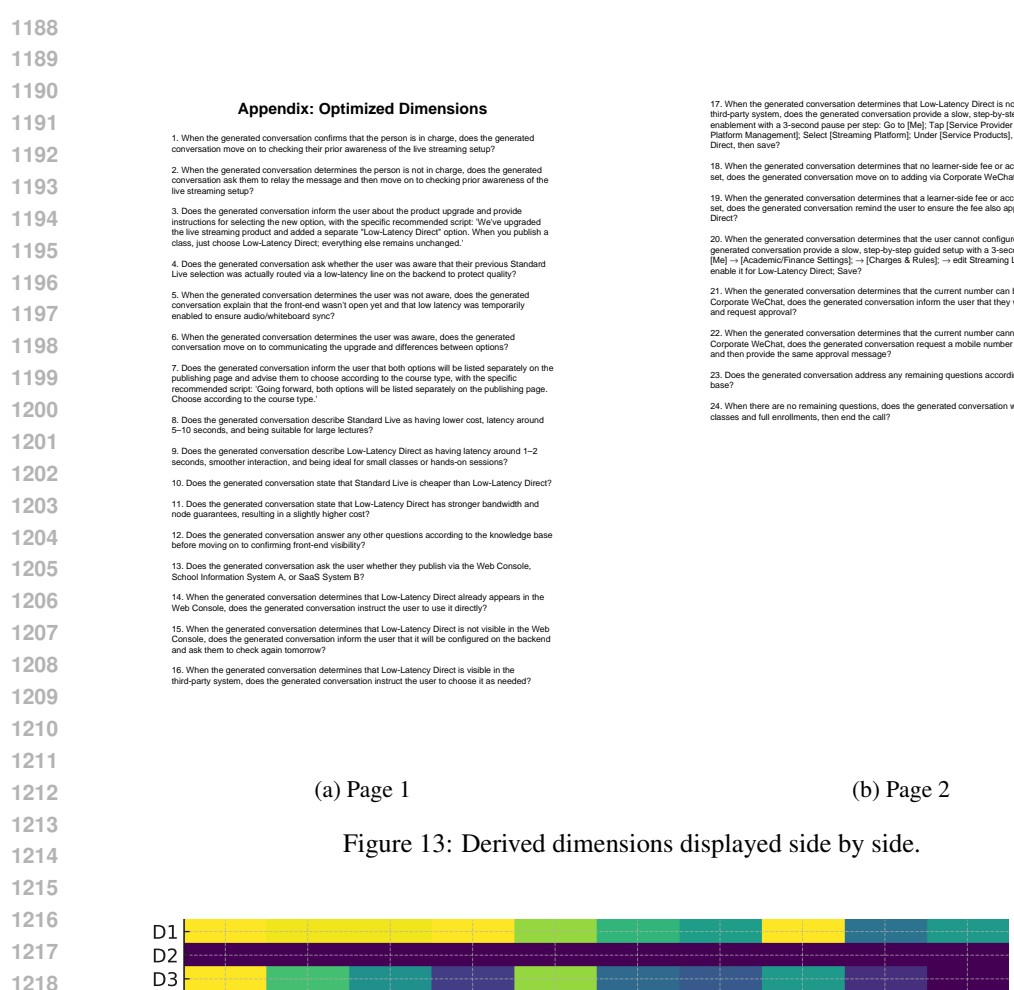

**Appendix: Optimized Dimensions**

1. When the generated conversation confirms that the person is in charge, does the generated conversation move on to checking their prior awareness of the live streaming setup?

2. When the generated conversation determines the person is not in charge, does the generated conversation ask them to relay the message and then move on to checking prior awareness of the live streaming setup?

3. Does the generated conversation inform the user about the product upgrade and provide instructions for selecting the new option, with the specific recommended script: 'We've upgraded the live streaming product and added a separate "Low-Latency Direct" option. When you publish a class, just choose Low-Latency Direct; everything else remains unchanged.'

4. Does the generated conversation ask whether the user was aware that their previous Standard Live selection was actually routed via a low-latency line on the backend to protect quality?

5. When the generated conversation determines the user was not aware, does the generated conversation explain that the front-end wasn't open yet and that low latency was temporarily enabled to ensure audio/whiteboard sync?

6. When the generated conversation determines the user was aware, does the generated conversation move on to communicating the upgrade and differences between options?

7. Does the generated conversation inform the user that both options will be listed separately on the publishing page and advise them to choose according to the course type, with the specific recommended script: 'Going forward, both options will be listed separately on the publishing page. Choose according to the course type.'

8. Does the generated conversation describe Standard Live as having lower cost, latency around 5–10 seconds, and being suitable for large lectures?

9. Does the generated conversation describe Low-Latency Direct as having latency around 1–2 seconds, smoother interaction, and being ideal for small classes or hands-on sessions?

10. Does the generated conversation state that Standard Live is cheaper than Low-Latency Direct?

11. Does the generated conversation state that Low-Latency Direct has stronger bandwidth and node guarantees, resulting in a slightly higher cost?

12. Does the generated conversation answer any other questions according to the knowledge base before moving on to confirming front-end visibility?

13. Does the generated conversation ask the user whether they publish via the Web Console, School Information System A, or SaaS System B?

14. When the generated conversation determines that Low-Latency Direct already appears in the Web Console, does the generated conversation instruct the user to use it directly?

15. When the generated conversation determines that Low-Latency Direct is not visible in the Web Console, does the generated conversation inform the user that it will be configured on the backend and ask them to check again tomorrow?

16. When the generated conversation determines that Low-Latency Direct is visible in the third-party system, does the generated conversation instruct the user to choose it as needed?

17. When the generated conversation determines that Low-Latency Direct is not visible in the third-party system, does the generated conversation provide a slow, step-by-step guided enablement with a 3-second pause per step: Go to [Me]; Tap [Service Provider / Streaming Platform Management]; Select [Streaming Platform]; Under [Service Products], check Low-Latency Direct, then save?

18. When the generated conversation determines that no learner-side fee or accelerated line fee is set, does the generated conversation move on to adding via Corporate WeChat?

19. When the generated conversation determines that a learner-side fee or accelerated line fee is set, does the generated conversation remind the user to ensure the fee also applies to Low-Latency Direct?

20. When the generated conversation determines that the user cannot configure the fee, does the generated conversation provide a slow, step-by-step guided setup with a 3-second pause per step: [Me] → [Academic/Finance Settings]; → [Charges & Rules]; → edit Streaming Line Surcharge and enable it for Low-Latency Direct; Save?

21. When the generated conversation determines that the current number can be added via Corporate WeChat, does the generated conversation inform the user that they will be added shortly and request approval?

22. When the generated conversation determines that the current number cannot be added via Corporate WeChat, does the generated conversation request a mobile number that can be added and then provide the same approval message?

23. Does the generated conversation address any remaining questions according to the knowledge base?

24. When there are no remaining questions, does the generated conversation wish the user smooth classes and full enrollments, then end the call?

(a) Page 1      (b) Page 2

Figure 13: Derived dimensions displayed side by side.

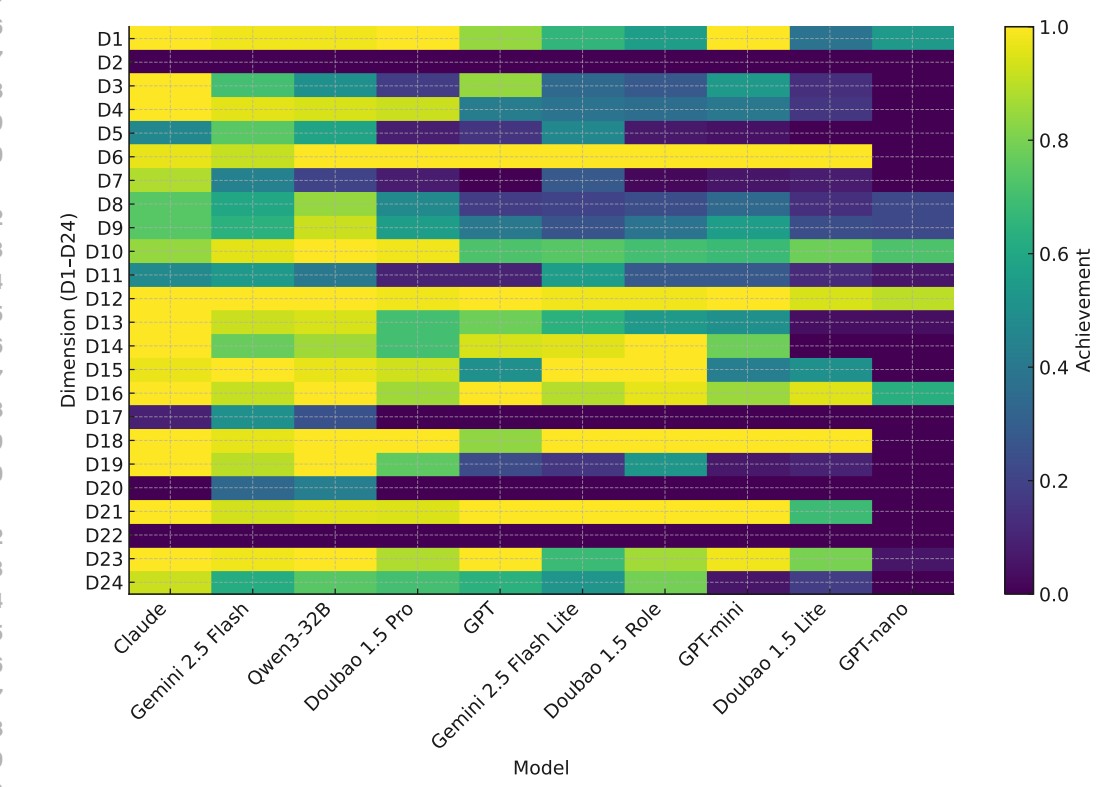

Figure 14: Per-dimension Achievement heatmap.

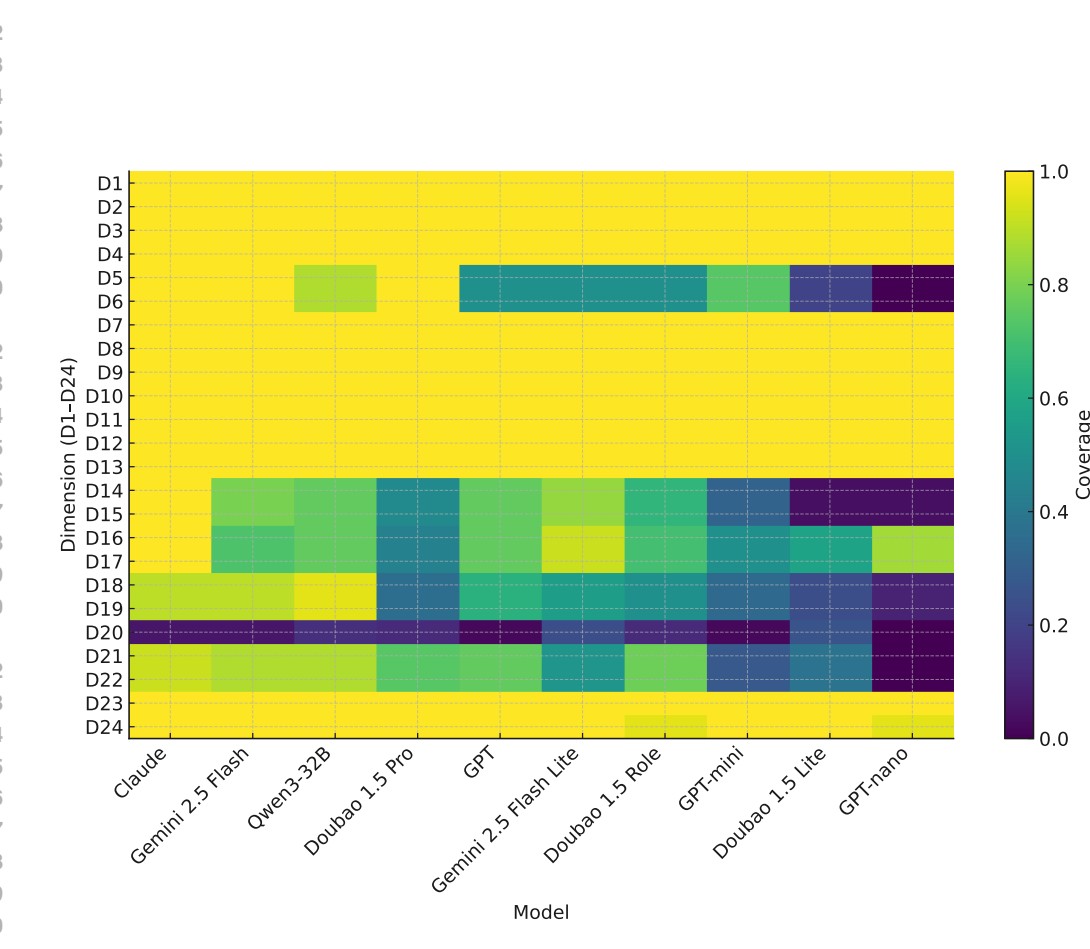

Figure 15: Per-dimension Coverage heatmap.

