# OpenReview forum: "One Instruction Is a Benchmark: End-to-End Instruction-Following Evaluation with FlexBench"
_ICLR.cc/2026/Conference — ICLR 2026 Conference Withdrawn Submission_

### Official Review · Reviewer_c4Co · 2025-10-16

**Soundness:** 2
**Presentation:** 1
**Contribution:** 2
**Rating:** 4
**Confidence:** 3

**Summary:**

This paper proposes FlexBench, a framework that automatically turns a single instruction into a complete benchmark with both a dataset and an evaluation suite. It’s basically about letting one instruction expand into a self-evolving evaluation process for instruction-following, which is both conceptually neat and practically useful.

**Strengths:**

The core idea—expanding one single instruction into a full dataset and evaluation pipeline—is genuinely interesting. It’s not only a clever conceptual move but also very practical. I can imagine LLM developers caring a lot about this kind of thing when debugging specific model behaviors, especially to ensure that their models perform well on rare but critical corner cases. The proposed paradigm could help make evaluation more targeted and meaningful in the real world, rather than relying on static, generic benchmarks.

**Weaknesses:**

+ The writing needs significant improvement. The paper is quite hard to follow at first read—I had to jump back and forth through notations and sections just to understand the setup and motivation behind each step. The clarity of exposition could be much better.
+ The biggest technical weakness is the lack of meta-evaluation—there’s no real check on whether the evaluation itself makes sense or is more reliable than existing alternatives. Most experiments validate component-level reliability or provide very indirect evidence. For example, Section 5.2 feels a bit circular.
+ The methodological assumptions are quite strong and not well justified. The framework assumes explicit, tree-structured “dimensions,” which might not exist cleanly in the real world. This human-designed prior could limit generalization. In practice, dimensions can be fuzzy or interdependent in ways that the current setup doesn’t model well.

**Questions:**

N/A

---

### Official Review · Reviewer_9ApZ · 2025-10-27

**Soundness:** 2
**Presentation:** 1
**Contribution:** 2
**Rating:** 2
**Confidence:** 4

**Summary:**

The paper is motivated by the observation that large language models (LLMs) vary widely in how well they follow specific, complex instructions. So static, general-purpose benchmarks fail to reflect real-world deployment needs. To address this, the authors propose FlexBench, which automatically builds a benchmark from a single instruction by first extracting a set of verifiable evaluation dimensions through a four-stage pipeline, then generating a conversation corpus using a user simulator. The simulator has an instruction-invariant backbone and adds only a minimal instruction-specific profile delta to prevent evaluation leakage. For evaluation, they introduce FlexEval, which aggregates tri-valued (yes/no/unknown) dimension-level judgments into two instruction-level metrics: Coverage (workflow progress) and Achievement (conditional correctness), using a gated aggregation rule that respects preconditions. The authors test their framework on 248 single-turn complex instructions and three multi-turn scenarios using ten leading LLMs. Results show Claude performing best overall, GPT-4.1 achieving high Coverage but lower Achievement, strong self-consistency across independently generated datasets, and near-perfect correlation between LLM-as-judge and expert annotations, confirming the method’s reliability and reproducibility.

**Strengths:**

The importance of synthetic data generation for LLM evaluation tasks is appreciated by the research community. The idea of generating an evaluation dataset from a single instruction is novel. The experiments have a great coverage of LLM models. The appendix gave details on the prompt templates used for the automated pipeline.

**Weaknesses:**

1. The entire framework's utility and the resulting benchmark's quality, diversity, and difficulty are critically dependent on the single seed instruction I. This is a major methodological vulnerability that is not sufficiently addressed. A poorly chosen I could result in a trivial, narrow, or ambiguous benchmark, leading to evaluation results that are not meaningful or generalizable. The main body of the paper lacks a discussion on how I should be selected, what constitutes a "good" seed instruction, and how sensitive the final evaluation scores are to variations in I.

2. The core pipeline (I → D(I), C(I)) is repeatedly described as LLM-driven (segmentation, criteria, verification, modification) and simulator-based, yet the paper never states which LLM(s) are used for dimension derivation, logic extraction, or simulation, nor does it justify those choices. This omission makes the benchmark hard to reproduce and to interpret (diversity, bias, and failure modes depend on the generator).

3. The paper uses verbose pseudo-formulas and heavy notation where a worked example would suffice (e.g., Figure 2 shows symbolic sums for Coverage/Achievement but gives no tight, illustrative walk-through). The exposition becomes harder to follow than necessary.

4. Appendix G states LLMs were “only to aid or polish writing,” which contradicts the LLM-driven pipeline described earlier. This creates confusion about what parts are automated by LLMs vs. scripted code.

5. The paper's strongest claims—near-perfect Pearson correlations with “expert annotations” (r=0.997/0.986)—are based on a limited validation set (72 dialogues). In addition, there's no information in main body or appendix on how the "expert annotations" are collected and how the selection process of "experts" was conducted.

6. While the framing is nice, constructing criteria from a single prompt and aggregating constraint-level decisions overlaps with existing benchmarks' themes; the paper’s novelty claims would need stronger empirical differentiation. Furthermore, generation from a single instruction is inherently a highly narrow evaluation set. The authors wanted to claim that their work mitigates the gap that "model’s 'general' ability is not a reliable proxy for its ability to follow a specific instruction" but there is no supporting claims on why generating a dataset from a single instruction is more reliable.

**Questions:**

1. You aim to avoid exposing evaluation dimensions to the simulator. How do you verify no indirect leakage occurs via shared prompts or persona fields? Any adversarial checks?

2. The paper mentions a “minimal, instruction-specific profile delta.” What quantitative bound defines “minimal,” and how does it influence difficulty?

3. The gating rule prevents “spurious credit,” but could it hide systematic base-action failures that mask branch difficulty?

4. You show high correlation with expert annotations and weak alignment with crowd labels. Which judge models were used, and are the same vendor families being judged—raising bias concerns?

5. Many headline comparisons (e.g., Claude best overall; GPT family scaling) are presented—are differences statistically significant across instructions and seeds?

6. How did you arrive at temperature 0.3? It doesn't avoid randomness during decoding.

---

### Official Review · Reviewer_Vpya · 2025-10-28

**Soundness:** 2
**Presentation:** 2
**Contribution:** 2
**Rating:** 2
**Confidence:** 3

**Summary:**

This paper describes a method to automatically evaluate the instruction following abilities of an LLM; given an instruction the system creates multiple segments (dimensions) for evaluating following and also creates multi-turn conversations based on the instruction to serve as input data for evaluation. The evaluation dimensions are first segmented into non-overlapping units of information (“segments”).  Then, each segment is converted into a dimension stated as an evaluation question. Conversations are generated using persona templates; each dialogue starts with the assistant issuing an opening line defined in the profile template. The simulator then generates user responses conditioned on the persona profiles. Models are evaluated against these instructions and conversational inputs. A human evaluation of the decomposition steps indicates that the results correlated with the LLM judges and that the automated methods tend to sometimes create finer-grained evaluation dimensions.

**Strengths:**

- Automated Instruction-following benchmark
- Scoring dimensions compare well/correlate with human annotations

**Weaknesses:**

- Paper is missing crucial details -- which LLM was used to create the segments, the conversations? How were sub-components of the pipeline validated (eg: persona requirements etc).
- No qualitative samples are included

**Questions:**

1. See Weakness - crucial details missing
2. It is not evident that this benchmark provides new insight into the instruction-following capabilities of LLMs. In comparison to established resources like ComplexBench (which explicitly classifies conjunctions, conditionals, multi-step constraints, etc.), it remains ambiguous what kinds of dimensions are actually studied. Are dimensions ever classified, or analyzed for coverage of logical and compositional properties? Table 2 simply lists aggregate statistics for “some” instructions and their corresponding dimensions, but does not enable a deeper understanding of the nature, variety, or complexity of the studied behaviors. If dimensions are bespoke per instruction but not categorized or typified, analytical potential is limited—how does this method aid systematic analysis?
3. What happens if prompts from established instruction-following (IF) benchmarks (e.g., InfoBench, ComplexBench) are processed by this system? Does the segmentation/dimension extraction reliably recover the same instruction components, or produce meaningful alignment or overlap? A discussion, or even empirical analysis, on compatibility or similarity with existing manual benchmarks would greatly clarify the strengths and limitations of the FlexBench pipeline.
4. How do results from the FlexBench evaluation correlate with model performance on established instruction-following or compositionality benchmarks? Are model scores predictive or consistent across benchmarks, or are there notable discrepancies? The paper would benefit from analysis or commentary addressing why FlexBench yields complementary, superior, or orthogonal insights, and how practitioners and researchers should interpret model performance using this benchmark/method in relation to other industry-standard benchmarks

---

### Official Review · Reviewer_8rto · 2025-10-31

**Soundness:** 2
**Presentation:** 2
**Contribution:** 2
**Rating:** 4
**Confidence:** 2

**Summary:**

This paper introduces FlexBench, a framework that automatically derives multiple distinct dimensions and a set of conversations for evaluation to form a benchmark from a single instruction. This paper also introduces FlexEval, which evaluate the conversations on each dimension and provide a tri-valued result, and aggregate these decisions into an instruction-level metric comprising coverage for workflow progress and achievement for conditional correctness. Experiments indicate the high quality of the decomposed dimensions compared with human-authored ones and the stability of the benchmark construction approach.

**Strengths:**

1. The FlexBench framework decomposes multiple dimensions from the original instruction, which enhances the comprehensiveness of the evaluation.
2. Experiments show that the results from two test sets generated from the same instruction by the framework maintain a high consistency, indicating the stability of the benchmark construction process.

**Weaknesses:**

1. The correlation between scores on each dimension and the capability of LLMs seems to be unclear. For example, there is a dimension called "Does the generated list provide an estimated cost for each date night idea?" as presented in Figure 2. What does it represent if a model get high score on this single dimension? If the dimension doesn't reflect a corresponding capability of models, the evaluation score of this single dimension will be meaningless.
2. The definition of the different settings V1, V2 and V3 in Figure 1 is incomplete. If they denote different settings of instruction length/type/form, the results in Figure 1 seem to reflect the models' capability discrepancy on different tasks, instead of the model-specific sensitivity to variations, which contradicts with the observation in the introduction.
3. There are several confusions in the presentation of the paper:
- In Section 3.2, there's no explanation for $\Delta(I)$ in the context.
- In Section 4.2, the signal "R" in equation (7), and the difference between $N$ and $\tilde{N}$ in quation (6) and (7) is not explained.
- In Figure 3, "converge" is not defined (seems like a typo of "coverage").
- In line 74-77, the citation format is mistakenly used (e.g. Qin et al. (2024) instead of (Qin et al., 2024)).

**Questions:**

1. The precondition groups mentioned in Section 4.1 seems not to appear in the subsequent contexts. How are they used in the FlexEval framework?
2. It is recommended to provide a specific case for the FlexBench and FlexEval framework to enhance clarity, such as the instruction and its corresponding dimensions and conversations, and how the conversation is then evaluated.

---

### Note · Authors · 2025-12-12

I have read and agree with the venue's withdrawal policy on behalf of myself and my co-authors.